# Advanced Biomimetic Multispectral Curved Compound Eye Camera for Aerial Multispectral Imaging in a Large Field of View

**DOI:** 10.3390/biomimetics8070556

**Published:** 2023-11-20

**Authors:** Yuanjie Zhang, Huangrong Xu, Yiming Liu, Xiaojun Zhou, Dengshan Wu, Weixing Yu

**Affiliations:** 1Key Laboratory of Spectral Imaging Technology, Xi’an Institute of Optics and Precision Mechanics, Chinese Academy of Sciences, No. 17, Xinxi Road, Xi’an 710119, China; zhangyuanjie17@mails.ucas.ac.cn (Y.Z.); hrxu4221@163.com (H.X.); liuyiming21@mails.ucas.ac.cn (Y.L.); zhouxiaojun@opt.ac.cn (X.Z.); wu.dengshan@opt.ac.cn (D.W.); 2Center of Materials Science and Optoelectronics Engineering, University of Chinese Academy of Sciences, Beijing 100049, China

**Keywords:** compound eye system, unmanned aerial vehicle, multispectral imaging, remote sensing

## Abstract

In this work, we demonstrated a new type of biomimetic multispectral curved compound eye camera (BM3C) inspired by insect compound eyes for aerial multispectral imaging in a large field of view. The proposed system exhibits a maximum field of view (FOV) of 120 degrees and seven-waveband multispectral images ranging from visible to near-infrared wavelengths. Pinhole imaging theory and the image registration method from feature detection are used to reconstruct the multispectral 3D data cube. An airborne imaging experiment is performed by assembling the BM3C on an unmanned aerial vehicle (UAV). As a result, radiation intensity curves of several objects are successfully obtained, and a land type classification is performed using the K-means method based on the aerial image as well. The developed BM3C is proven to have the capability for large FOV aerial multispectral imaging and shows great potential applications for distant detecting based on aerial imaging.

## 1. Introduction

Unmanned aerial vehicles (UAV) are commonly used in multispectral remote imaging, with the advantages of a low price, easy use and flexible mobility [1]. Among the multiple kinds of UAVs, quadrotor and six-rotor UAVs are widely applied in aerial platforms for multispectral snap-shot imaging. In general, these UAVs have a payload capacity under 7 kg, a speed of 12 m/s and a flight ceiling altitude of 500 m [2]. There are different kinds of products for airborne multispectral imaging, like ADC lite (TetracamInc., Chatsworth, CA, USA), RedEdge (Micasensen Inc., Seattle, WA, USA), Parrot Sequoia (Parrot Inc., Paris, France), etc.

UAV-mounted multispectral cameras are supposed to be small and light and have a large FOV, which fits well with the features of artificial compound eye (ACE) systems. ACE is a kind of biomimetic imaging system inspired by the compound eye structure of insects. The compound eye in nature contains multiple ommatidia arranged on a free-shaped eyeball, and each of the ommatidia has its own photoreceptor. The structure allows for imaging independently and achieves a large FOV with a compact size. 

The ommatidia of ACE are usually designed as separate imaging channels, whose optical system is optimized according to the software for aberration correction. By now, many research works about ACE systems have been reported in order to take advantage of its features. In general, compound eye imaging systems include planar and curved compound eye systems. Based on the Thin Observation Module by Bound Optics (TOMBO) system proposed in 2000 [3], the planar compound eye system contains two major parts: a planar lens array and an image sensor. These systems achieve high-resolution, high-speed imaging and 3D information acquisition [4,5,6,7]. The curved compound eye system contains a curved lens array with a planar or curved image sensor, whose structure is closer to the insect compound eyes in nature. In the research, the curved compound eye system shows the advantage of ultra-large FOV imaging with low distortion. The application of curved systems mainly focuses on shape detecting, trajectory tracking, orientation detecting and 3D measurement [8,9,10,11,12,13].

As each ommatidium works separately in a compound eye, the insects have been found to specialize different parts of the compound eye for different utilities [14,15]. This discovery inspires the development of multispectral ACE systems. Most of the multispectral ACE systems are based on the TOMBO system and the color TOMBO system in 2003 [16]. Following the work of Tanida’s group, Shogenji et al. proposed a multispectral imaging system based on the TOMBO structure by integrating narrow-band filters to ommatidia lenses [17]. In 2008, Scott A. Mathews et al. proposed a developed TOMBO multispectral imaging system with 18 lens imaging units with filters [18]. In 2010, Kagawa et al. reported a multispectral imaging system on the basis of the TOMBO structure that utilizes the rolling shutter CMOS to achieve high-speed multispectral imaging [19]. Other similar multispectral imaging systems are widely proposed for different applications [20,21,22].

The multispectral ACE system utilized on the UAV platform was first achieved in 2020 by Nakanshi et al. from Osaka University [23]. In the article, a prototype with an embedded computer and a multispectral TOMBO imaging system was demonstrated as shown in Figure 1. Multispectral images of a ground oil furnace and vehicle target were captured with the system at a flight altitude of 3–7 m and were also analyzed with the normalized vegetation index (NDVI). This work shows the capacity of the multispectral ACE system for aerial imaging. 

In previous studies, we proposed a so-called Multispectral Curved Compound Eye Camera (MCCEC) in 2020 [24]. The system was designed to be able to achieve an FOV of 120 degrees with seven-wavebands multispectral imaging, with a rather short focal length of 0.4 mm. Based on the design, a biomimetic multispectral curved compound eye camera (BMCCEC) prototype was demonstrated in 2021. The BMCCEC system achieved a maximum FOV of 98 degrees and an improved focal length of 5 mm [25]. However, the structure design of the prototype causes a blind area and lacks mounting space for filters, which is limited in the prototype aerial imaging.

To solve these issues of the BMCCEC system and explore the possibility of curved ACE systems in aerial imaging, an advanced BMCCEC (BM3C) system is proposed in this article. Based on design theory analysis, the system achieves multispectral imaging without a blind area, with a maximum FOV of 127.4 degrees, a focal length of 2.76 mm and seven multispectral channels. To test the performance of the system, spectral calibration and correction, an airborne imaging experiment and data processing are performed with the prototype.

## 2. Multispectral Imaging System

### 2.1. System Design Theory

The BM3C system achieves multispectral imaging based on clusters of multispectral ommatidia. Thus, the arrangement of the ommatidia, FOV and focal length are solved theoretically to avoid a blind area and image overlap in this work. 

#### 2.1.1. The Arrangement of the Multispectral Lens Array

ACE systems contain multiple methods of lens arrangement. To achieve multispectral imaging, the included angle between the neighbor channels should be as consistent as possible to avoid wasting the FOV. Based on the tessellation rule, the most common arrangements are regular triangles and quadrilaterals. Three possible arrangements are shown in Figure 2. It can be seen that the arrangement based on regular triangles has a consistent distance between nodes, as well as the maximum number of spectrum channels without FOV waste.

Thus, an arrangement method based on triangle mesh is proposed, as shown in Figure 3a. As the highlighted one shows, each cluster unit contains seven spectral channels. The mesh is then projected to a curved shell to determine the actual ommatidia arrangement, as shown in Figure 3b.

To determine the central wavelengths of seven channels, a survey of commercial aerial multispectral imaging systems is made. Representative products contain imaging channels of blue, green, red and NIR wavebands for agriculture and ground object classification applications. Therefore, the working waveband of the system is chosen as 440 nm–800 nm. For the prototype system, seven central wavelengths are designed as 500, 560, 600, 650, 700, 750 and 800 nm to acquire the critical spectral information of ground objects with green leaves, and this working waveband can cover the so called ‘red edge’ reflection spectrum of the green leaves [26].

#### 2.1.2. The FOV

In the design of BM3C, 169 ommatidia are arranged on the sphere structure with the sphere projection method [25]. While the included angle between the light axes of two neighbor ommatidia channels is seven degrees, the system should achieve a maximum FOV of 120 degrees and a maximum multispectral imaging field of 98 degrees.

To avoid a blind area, the FOV of each ommatidium is calculated with an overlap calculation equation [27], as shown in Figure 4.
(1)$$d_{c}=\frac{d\cdot \sin\frac{\beta }{2}}{\tan\alpha }=\frac{[R\cdot \cos\frac{\beta }{2}+R\cdot \sin\frac{\beta }{2}\cdot \cos(\alpha \;-\;\frac{\beta }{2})/\sin(\alpha \;-\;\frac{\beta }{2})]\cdot \sin\frac{\beta }{2}}{\tan\alpha }$$
where the radius of the curved shell *R* is 125 mm, *α* represents the half FOV of each channel and *β* represents the included angle between the light axes of a pair of neighbored multispectral channels of the same waveband. By solving the included angle in the 3D model of ommatidia distribution, the maximum angle *β* is 18.52 degrees according to the result. Considering the working environment of the airborne camera, the minimum working distance is set as 1000 mm. Under this condition, the minimum FOV of the ommatidium is solved as ±10.57 degrees. The FOV is set as ±12 degrees during the design procedure to leave space for machining and adjustment allowance.

#### 2.1.3. The Focal Length

To avoid the overlap between neighboring sub-images, there should be enough of a physical interval between the neighbor ommatidia. According to a previous study, there should be a minimum interval of 0.4 mm on the imaging plane. Thus, the focal length *f* of the whole optical system can be solved with Equation (2).
(2)$$\begin{array}{l} r=\frac{s_{p}\;\times \;n_{p}\;-\;n_{f}\;\times \;l_{f}}{2\;\times \;n_{o}}\\ f=\frac{r}{\tan\theta } \end{array}$$
where $s_{p}$ and $n_{p}$ represent the size and number of the pixels of the CMOS sensor. $n_{f}$ = 14 and $l_{f}$ = 0.4 mm represent the number and length of the interval between the image of neighbor ommatidia. $n_{o}$ = 15 is the number of the ommatidia on the diagonal of the hexagon array. *θ* = 12 degrees is half of the FOV of each ommatidium channel. From Equation (2), the focal length *f* of the system is solved as 2.7 mm. The focal length *f* is distributed into the two optical subsystems with the Newton formula.

### 2.2. Prototype of BM3C

As the critical parameters are solved with theory analysis, a prototype of BM3C is designed. In the optical design process, the system is optimized with optical design software. Figure 5 shows the 3D model of the system, as well as two critical parameters from the simulation, the Modulation Transfer Function (MTF) and the distortion of the edge channel of the system. The simulation system has an MTF larger than 0.36 at 55 lp/mm and a distortion less than 2% in all channels, which represents a fine imaging quality.

According to the design result, a camera prototype is manufactured, as shown in Figure 6, which contains three parts.

A curved shell that contains 169 multispectral ommatidia; each ommatidium is a doublet lens with a focal length of 17.02 mm. Seven groups of narrowband filters are mounted before the ommatidia. The whole shell contains 25 filters with a nominal central wavelength of 500 nm and 6 groups of 24 filters with nominal central wavelengths at 560, 600, 650, 700, 750 and 800 nm; the arrangement of the filters is shown in Figure 2.An optical relay system with eight glass lenses, which transforms the curved image plane of the ommatidia array into a planar one. According to the Newton formula, the focal length is designed as 1 mm to achieve a whole system focal length of 2.7 mm.An Imperx Cheetah C5180M CMOS camera as the image sensor, with a resolution of 5120 × 5120 and a pixel size of 4.5 μm.

Several performance tests are conducted based on the BM3C system. The important parameters of the camera are listed in Table 1.

## 3. Image Reconstruction Method

### 3.1. Method Based on Reprojection and Feature Detection

#### 3.1.1. Error Analysis of the Image Reprojection

In the former study [25], a calibration-free image reconstruction method was proposed, as shown in Figure 7a. The method may introduce errors to the image reconstruction procedure in two major variables, the *x* and *y* coordinates of the principal point and the focal length of the system. Figure 7b shows an example of the error introduction. The reprojection method causes a large image registration error and leads to inaccurate multispectral measurement results.

The error introduced by the image reconstruction method can be described with Equation (3).
(3)$$\begin{array}{l} \Delta_{\mathrm{yo}}=\frac{\Delta_{o}(d\;+\;f)}{f}\\ \Delta_{\mathrm{yf}}=\frac{\Delta_{f}\mathrm{dx}}{f\;(f\;-\;\Delta_{f})} \end{array}$$
where f represents the theoretical focal length of the channel, O represents the theoretical coordinate of the principal point, $\Delta_{f}$ and $\Delta_{O}$ represent the error of the focal length and principal point and $\Delta_{\mathrm{yf}}$ and $\Delta_{\mathrm{yo}}$ represent the errors caused by the introduced errors of two variables. According to Equation (3), the reconstruction error caused by the principal point only relates to $\Delta_{O}$, and the reconstruction error caused by the focal length has a positive correlation with the distance x between the image point and principal point. Thus, to reduce the reconstruction error, an image reconstruction method combined with feature detection and image registration is proposed.

#### 3.1.2. The Combined Reconstruction Method

The image reconstruction method may lead to an error from the position of the image pixel. A new reconstruction method combines feature detection and the reprojection method to reduce the error of image reconstruction. This combined image reconstruction method is shown in Figure 8 and contains three main procedures:

First, the positions of all sub-images of ommatidia are found on the image plane with a generated and adjusted hexagon grid. As the center and radius of each circle image are determined, 127 multispectral clusters are noted by looking at the list of ommatidia and searching for the six nearest neighbor ommatidia for each ommatidium, except for the ones on the edge.After the multispectral clusters are noted, the image registration is performed based on the SIFT (Scale-Invariant Feature Transform) feature extraction algorithm, and the homography matrixes of the six surrounding channels in relation to the main channel are acquired with feature matching and the mismatching is reduced with the RANSAC (RANdom SAmple Consensus) algorithm [28]. The thresholds of the algorithm are manually adjusted to achieve the best matching and registration results.With the homography matrix and the image registration method, the projected positions of each pixel in the valid imaging area in the central ommatidium of each cluster are calculated and noted in a look-up table to achieve real-time multispectral image reconstruction.

The reconstructed image of a multispectral cluster unit is shown in Figure 9a, and Figure 9b shows the overlap map of the cluster. In Figure 9b, the white part in the center represents the valid imaging area of seven wavebands, and the surrounding areas represent the areas where fewer than seven wavebands overlap. 

Compared with the reprojection method, the combined method achieves a lower registration error of fewer than two pixels at the same place. Figure 10 shows a feature point in the image and two spectral channels with reconstruction deviation, and the reprojection method shows a larger deviation. Thus, the combined reconstruction method leads to a multispectral image acquisition with a higher precision, and the look-up table makes the program more than four times faster.

#### 3.1.3. Working Distance

As a compound eye system, the baseline and included angle between the light axis of ommatidia generate optical parallax. On one hand, this provides the capability to achieve distance measurement and 3D information acquisition; on the other hand, it causes image mismatching while using a look-up table in a short distance. Thus, the relationship between the parallax and the object distance has to be evaluated to determine the working distance of the system.

The imaging parallax of a pair of neighbor ommatidia is shown in Figure 11:

which can be solved with Equation (4):(4)$$\begin{array}{l} b=R\tan\theta \\ c=d\;-\;\frac{R}{\cos\theta }\;+\;R\\ a=\sqrt{b^{2}\;+\;c^{2}\;-\;2\cdot b\cdot c\cos(90\;+\;\theta )}\\ x=f\cdot \tan\{90\;-\;\arcsin[\frac{c\cdot \sin(90\;+\;\theta )}{a}]\}\\ res=x\;+\;x’ \end{array}$$
where the imaging procedure of an ommatidium channel is simplified as a pinhole camera model. *θ* is the included angle between the light axis OA and the object position OB, while the neighbor ommatidium has an included angle of 7 − *θ*. Both angles are 3.5 degrees for simplification. *d* is the distance of the object point. *f* = 2.7 mm is the focal length of the optical system. And *R* = 125 mm is the radius of the entrance pupil sphere. The parallax Res is a function relating to the distance d.

To avoid the mismatching caused by the parallax, the relationship between the object distance and parallax is solved, as shown in Figure 12.

The variation in the parallax caused by the system remains fewer than 0.1 pixels when the object distance is large enough, which can be assumed to be invariant in image reconstruction. Thus, the parallax with an object distance of 100 m is chosen as the static value, and the object distance of the pre-matching target should be larger than 8.4 m. 

### 3.2. Method of Multispectral Information Acquisition

#### 3.2.1. Radiometric Correction Based on Calibration

As described, the multispectral information of a single target is reconstructed from the image captured by a cluster of multispectral ommatidia. To invert the original radiation characteristics of the objects, spectral calibration and radiometric calibration are performed with the BM3C.

The spectral calibration applies the monochromator as the light source to measure the central wavelength and full-width-at-half-maximum (FWHM) of each spectrum channel. The monochromator works in the scanning mode, the wavelength scan range is 475–825 nm and the step is 2 nm. Figure 13 shows a set of images captured by an ommatidium during scanning. By extracting the Digital Number (DN) value of each pixel and a fitting procedure with a gaussian function, the central wavelength and half-band width of the spectrum channels are solved.

During the spectral calibration, the result of the central wavelengths of different sample points shows a nonuniform result. The nonuniformity should be caused by the feature of the filter based on the F-P structure, which makes the central wavelength shift to shortwave while the incident angle grows larger. To obtain more accurate calibration parameters, the image is divided into multiple rings according to the radius of the certain pixel, and the central wavelength of each ring is solved separately. The result of five rings is shown in Table 2. The deviation caused by the filter is about 3–5 nm from the center of the image to the edge.

The radiometric calibration is performed in a dark room, with an integrating sphere as the light source and a calibrated ASD spectrometer as the standard radiation reference. The aperture of the camera and the fiber sensor head of the spectrometer are placed in the integrating sphere. The image of the camera and the light radiation intensity curve measured by the spectrometer are recorded correspondingly as the working current of the integrating sphere is adjusted. As the central wavelength differs in rings, the radiometric calibration coefficient is solved with rings as the unit. Radiometric correction is achieved with Equation (5).
(5)$$L=\alpha_{ij}DN_{avgi}+\beta_{ij}=\alpha_{ij}(\alpha_{k}DN_{k}+\beta_{k})+\beta_{ij}$$
where $\alpha_{ij}$ and $\beta_{ij}$ are the absolute radiometric correction coefficients of ring *i* and wavelength *j* and $\alpha_{k}$ and $\beta_{k}$ are the relative radiometric correction coefficients of pixel *k*. With the data recorded in the integrating sphere experiment, all coefficients in Equation (5) are solved with a curve fitting tool.

#### 3.2.2. Multispectral Information Acquisition

With the reconstruction method and system calibration, the original radiation intensity curve of the whole scene can be reconstructed from the raw image captured by the BM3C. The multispectral information acquisition procedure of a chosen pixel on the reconstructed image involves the following steps:By solving the included angle of the chosen object pixel and the light axis of each ommatidium channel, the nearest channel is picked out, and the image pixel of the object is solved according to the projection theory.By checking the look-up table via the combined reconstruction method, the corresponding pixels in the other six images of the multispectral cluster are found.With the spectral and radiometric calibration data, the central wavelength and the correction coefficient of all points are determined. The radiation intensity curve is solved using Equation (5).

With the solved radiation intensity curve, the spectral features of the captured ground objects can be read and analyzed. As BMCCEC can only solve the reflection curve of an area due to the blind area, the BM3C system can record the multispectral information of all pixels in the large object scene and acquire an absolute radiation intensity curve.

## 4. Airborne Imaging Experiment

To test the imaging capability of the BM3C and the reconstruction method, an airborne imaging experiment was performed. During the experiment, a customized six-rotor UAV with a 12 V power supply for the camera and a programmable onboard computer was used as the boarding platform of the camera. The BM3C was mounted below the UAV with a customized fixture, and the lens was aimed to the ground for remote sensing, as shown in Figure 14.

The onboard computer was connected to the flight control system of the UAV with the DJI onboard SDK and contained a recording program. The system was programmed to start recording with a set delay after detecting the take-off signal from the UAV and to stop recording when detecting the landing signal.

The imaging experiment is in the Xi’an Institute of Optics and Mechanics. During the recording procedure in the experiment, the exposure time of the camera was set to 3000 μs, and the acquisition framerate was set to 2 fps. According to the data of the flight control system, the camera recorded at a maximum altitude of 75 m, and the horizontal flight range was 100 m.

## 5. Results

In the airborne imaging experiment, 795 images were recorded during a flight procedure. The images are reconstructed with the method described in Section 3. Figure 15 shows a series of multispectral images reconstructed from one of the raw images. 

The inverted radiation of four typical ground objects is sampled to evaluate the imaging result. In Figure 16, the absolute radiation intensity curves of four typical ground objects are picked from an image at a higher altitude. As shown in Figure 16b, the curve of red and green plants shows the typical response curve of the plant, which has an absorption band of around 600 nm to 650 nm and a high reflection at the waveband ranging from 750 to 800 nm, confirming the different intensities of 500–600 nm. Also, the curves of the same kind of target in a different area of the FOV show a high uniformity.

The spectrum curves of the test points in Figure 16 are reasonable and accord with the reflection characteristics of the objects. An object classification based on the k-means clustering algorithm is performed to clearly show the application potential. The result of the object classification is shown in Figure 17.

As can be seen clearly, the ground objects are classified into three categories: plants, a high-reflectivity concrete/building and a low-reflectivity concrete/wet road. Therefore, the prototype BM3C and the reconstruction method show the capacity for ground target recognition. The NDVI of the scene can be solved with the clustering result. With training data and a convolutional neural network, more detailed classification can be performed with the camera.

## 6. Discussion

In summary, aerial multispectral imaging with a prototype BM3C was demonstrated in this work. The camera contains 169 ommatidia distributed on a curved structure and achieves a maximum FOV of 120 degrees and a maximum seven-wavelength multispectral imaging FOV of above 98 degrees. Compared with the former BMCCEC prototype, the camera proposed in this article achieves the multispectral imaging of all pixels in the FOV with no blind area in the image (Figure 18).

A comparison of the BM3C prototype and the state-of-the-art aerial multispectral cameras is listed in Table 3 In comparison with previous research, our BM3C has a unique curved compound eye structure with multispectral imaging ability, preserving the larger FOV and more multispectral wavebands at the same time. However, the prototype system also shows a larger size and weight, which can be further reduced by changing the structure material and using a more compact design.

A comparison of the BM3C prototype and the state-of-the-art compound eye multispectral systems is listed in Table 4. With the unique multispectral lens array arrangement and the optical relay sub-system design, BM3C can achieve a large FOV by spreading out cluster units on a curved shell and a high resolution by optical optimization. Thus, the BM3C shows apparent advantages, especially in terms of the field of view, the focal length, the imaging resolution and the number of multispectral wavebands in comparison with others.

According to the results of this article, future study of the system can be undertaken in the following areas: Calibration of a large number of imaging channels. In the compound eye imaging system, normal calibration methods for array cameras perform relatively poorly because of two reasons: the low imaging resolution of each ommatidium and the hardship in the parameter optimization from the large number of the imaging channels. An appropriate calibration method for the system should give rise to a better image registration result.Multidimension information sensing. As the compound eye imaging system shows the capability of multispectral imaging, the system can also be used for multidimension information sensing. For example, by attaching a polarizer with different polarization angles to the ommatidia, the system can capture the polarization information of different polarization angles for navigation. This may bring on many new applications of the compound eye system.Lightening and miniaturizing the camera. As shown in Table 3, the BM3C achieves a large FOV and more spectrum channels than common aerial multispectral cameras but a larger volume and weight. In the future, the system will be lightened and miniaturized via multiple ways, including using lighter materials like resin for lenses and carbon fiber for the shell and using aspheric lenses for a more compact optical system.System application. Like other multispectral imaging devices, BM3C can be applied to fields like the parameters analyzing of crops or vegetation and the researching of biomass or biocommunities in a large area.

## Figures and Tables

**Figure 1 biomimetics-08-00556-f001:**
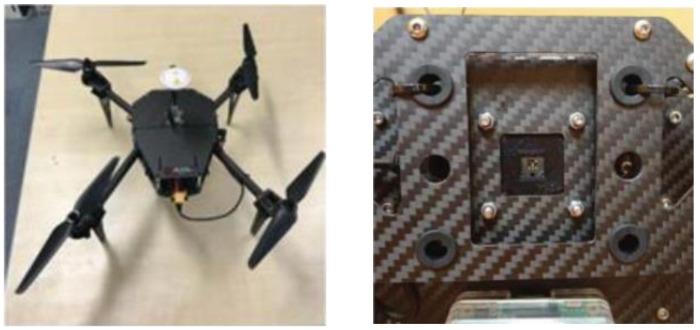
The UAV and multispectral TOMBO system by Nakanshi et al. [23].

**Figure 2 biomimetics-08-00556-f002:**
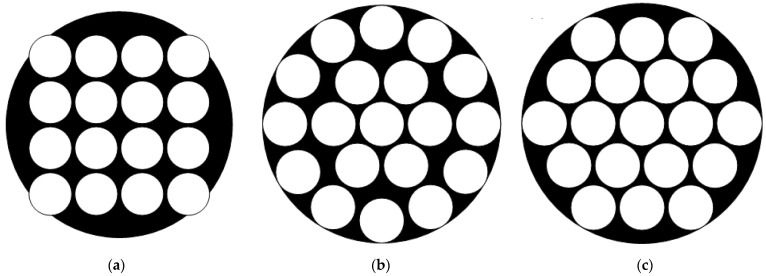
Arrangement methods of the compound eye system: (**a**) quadrilaterals, (**b**) circles, (**c**) triangles.

**Figure 3 biomimetics-08-00556-f003:**
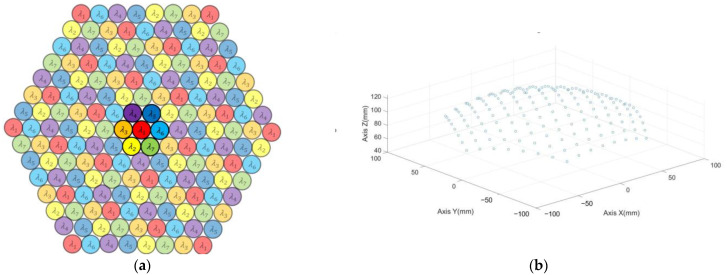
The arrangement of the multispectral filters: (**a**) spectrum arrangement, (**b**) coordinate arrangement.

**Figure 4 biomimetics-08-00556-f004:**
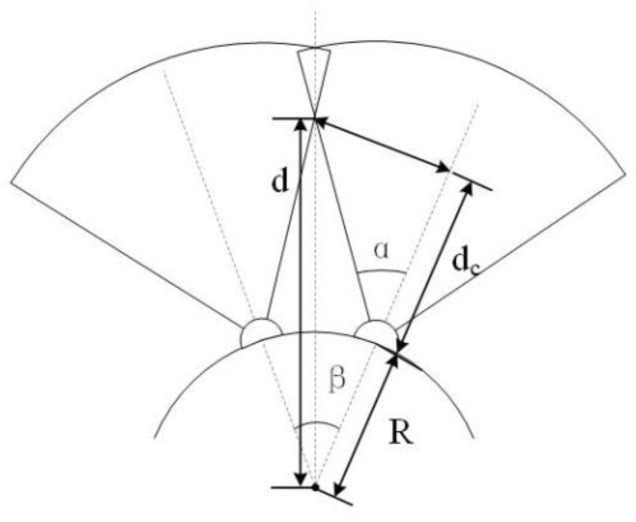
The FOV overlap of neighbor ommatidia.

**Figure 5 biomimetics-08-00556-f005:**
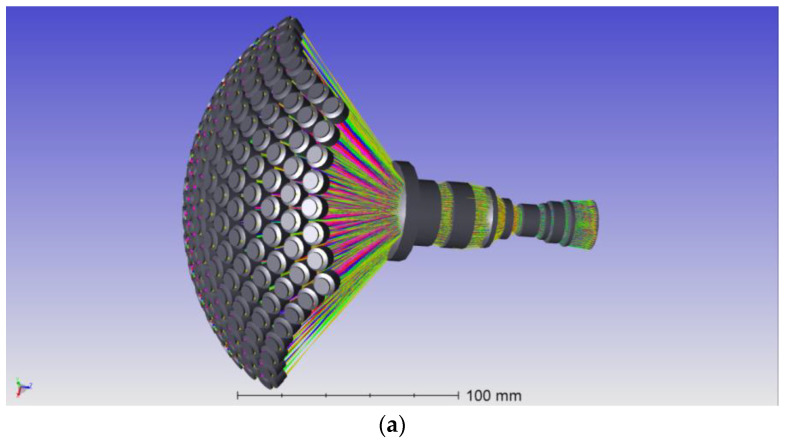
Simulation result of BM3C. (**a**) The 3D model of BM3C’s optical system, (**b**) the MTF simulation result of the edge channel, (**c**) the distortion simulation result of the edge channel.

**Figure 6 biomimetics-08-00556-f006:**
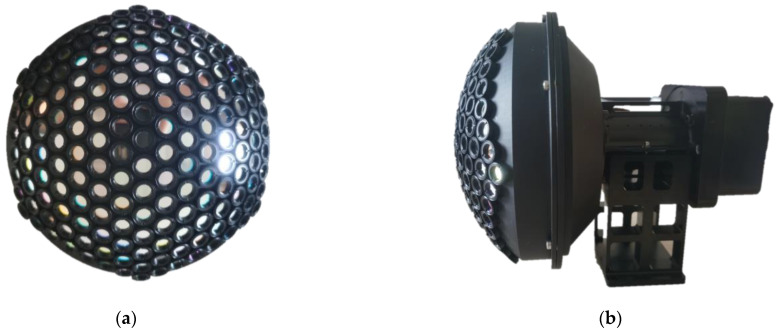
(**a**) The front-view multispectral array and (**b**) the side view of the prototype BM3C system.

**Figure 7 biomimetics-08-00556-f007:**
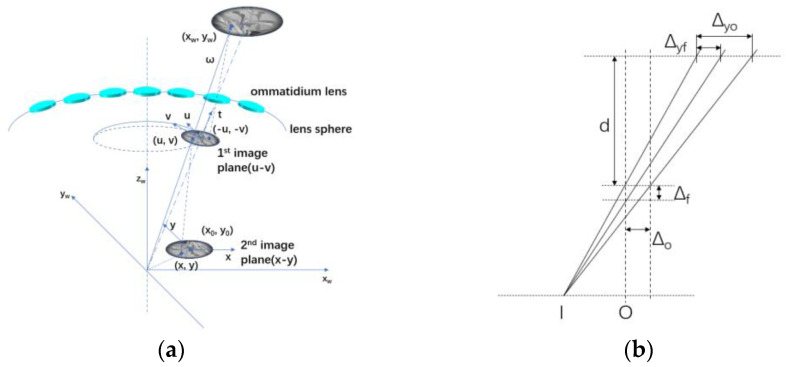
The calibration-free image reconstruction method. (**a**) The coordinates map, (**b**) the error analysis theory.

**Figure 8 biomimetics-08-00556-f008:**
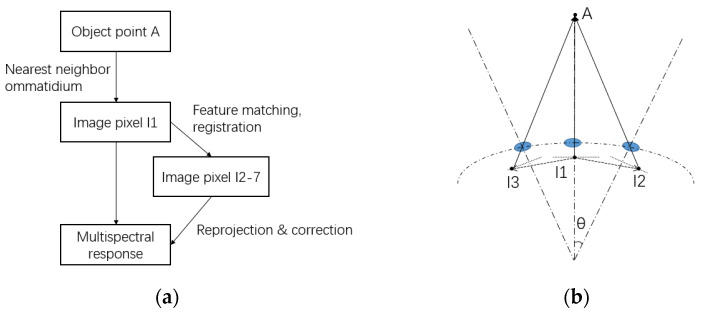
The flow chart and schematic diagram of the combined image reconstruction method. (**a**) Flow chart of the method, (**b**) the schematic diagram.

**Figure 9 biomimetics-08-00556-f009:**
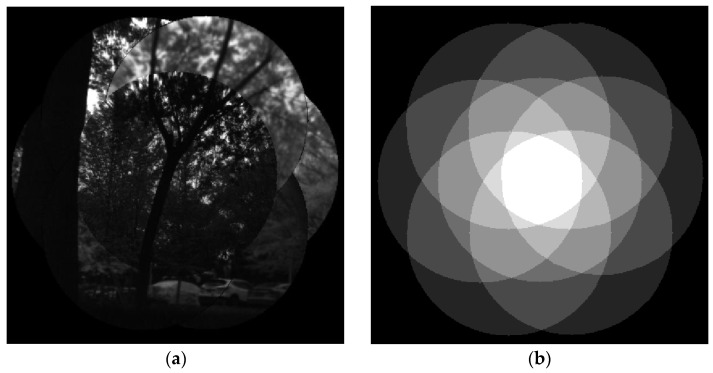
The registration image of a multispectral cluster unit. (**a**) The original image (**b**) overlap map of the cluster.

**Figure 10 biomimetics-08-00556-f010:**
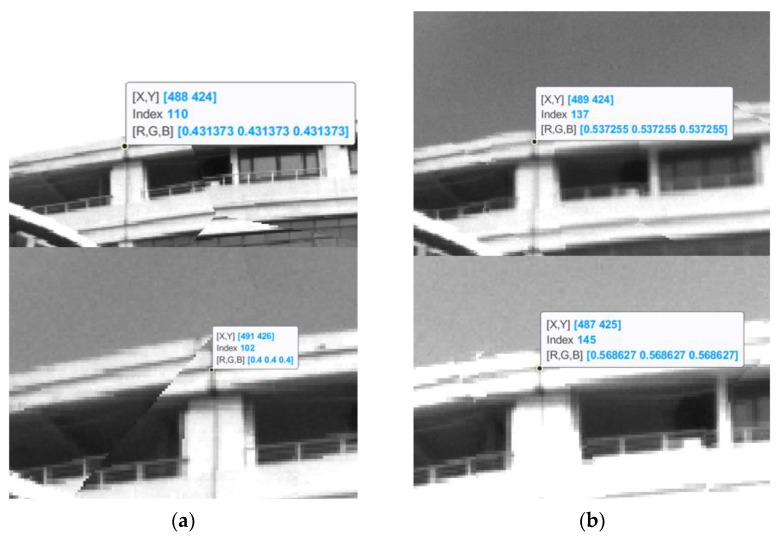
The comparison of two reconstruction methods. (**a**) The reprojection method, (**b**) the combined method.

**Figure 11 biomimetics-08-00556-f011:**
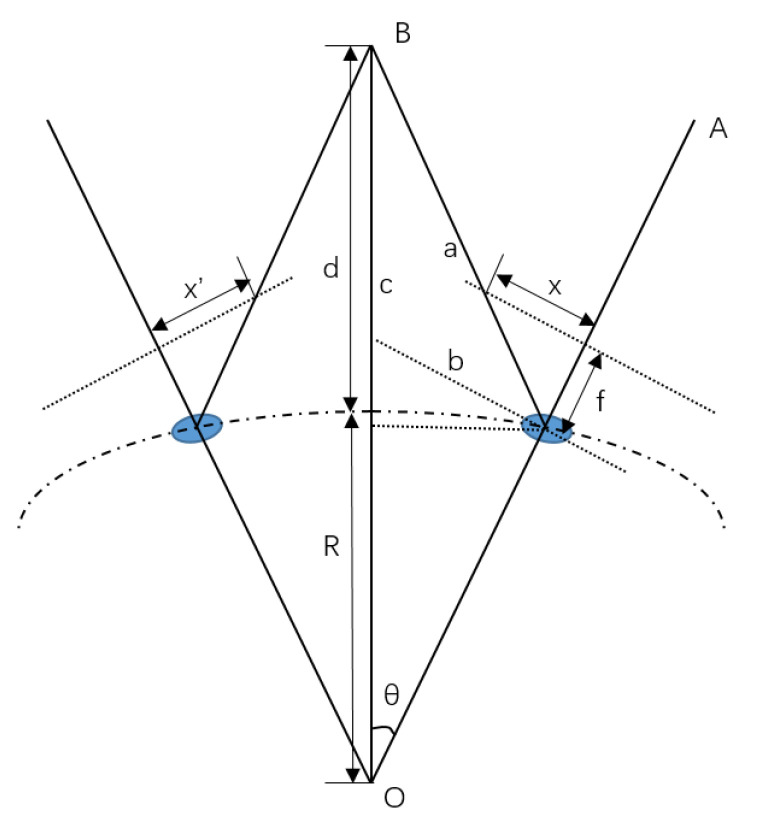
Imaging parallax of a pair of ommatidia.

**Figure 12 biomimetics-08-00556-f012:**
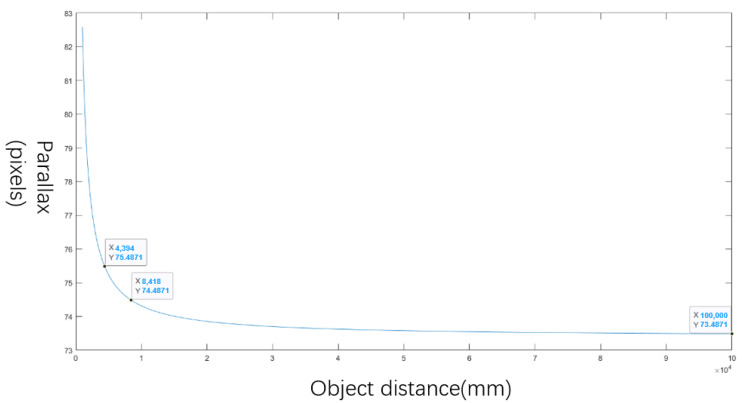
The relationship between the object distance and parallax.

**Figure 13 biomimetics-08-00556-f013:**
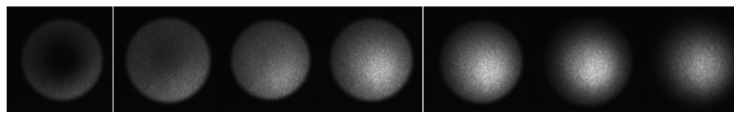
The captured image series during the spectral calibration procedure.

**Figure 14 biomimetics-08-00556-f014:**
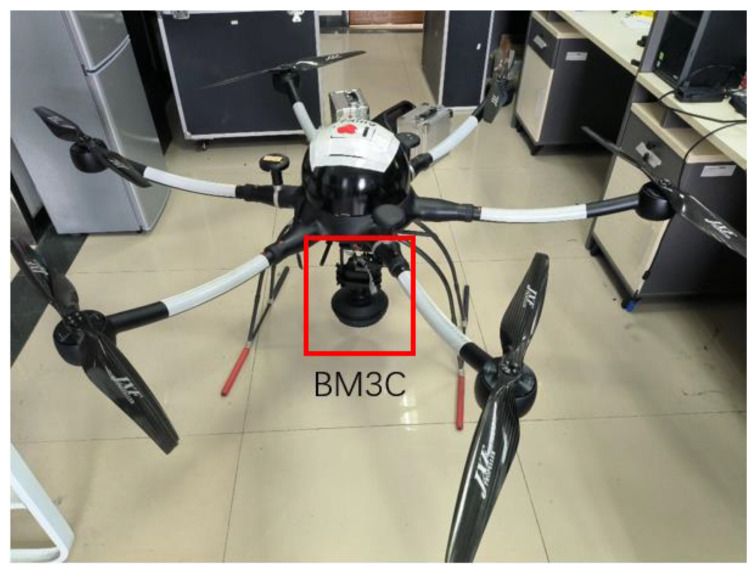
The UAV used in the experiment and the mounted camera.

**Figure 15 biomimetics-08-00556-f015:**
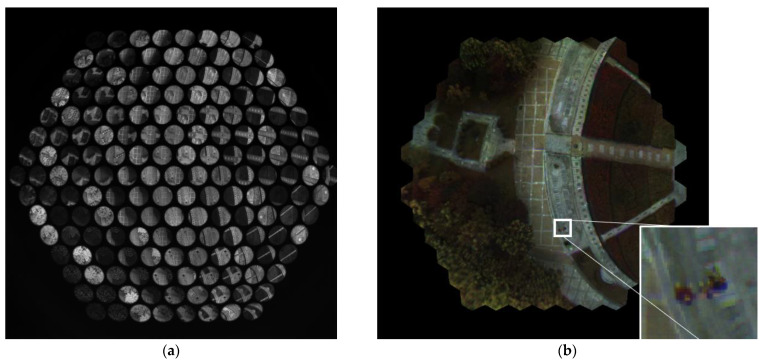
Reconstructed image of seven spectral channels of one scene. (**a**) The raw image, (**b**) reconstructed pseudo color image with enlarged people targets, (**c**) reconstructed image of 506, 560, 602, 653, 704, 751 and 801 nm wavelengths.

**Figure 16 biomimetics-08-00556-f016:**
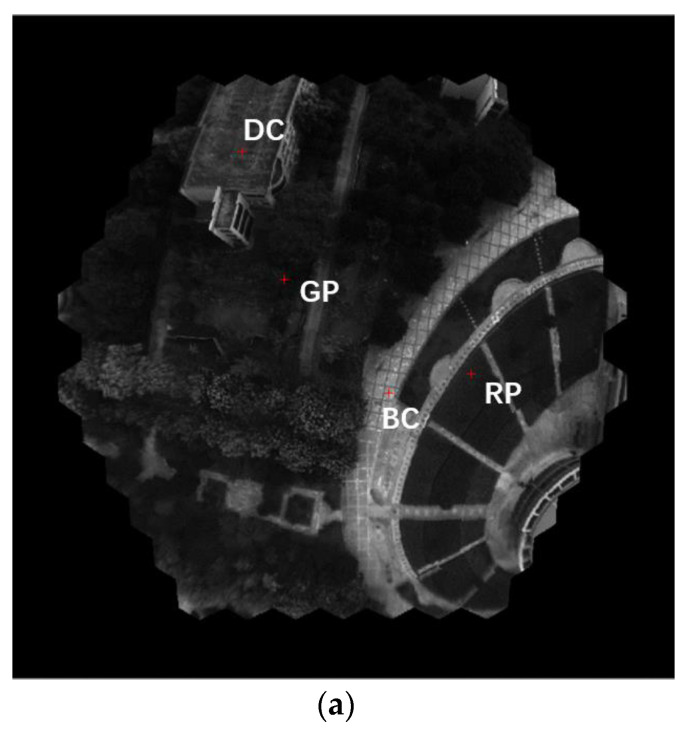
The radiation intensity curves of typical targets. (**a**) Reconstructed image of 560 nm with sample points, (**b**) curves of plant targets, (**c**) curves of concrete targets.

**Figure 17 biomimetics-08-00556-f017:**
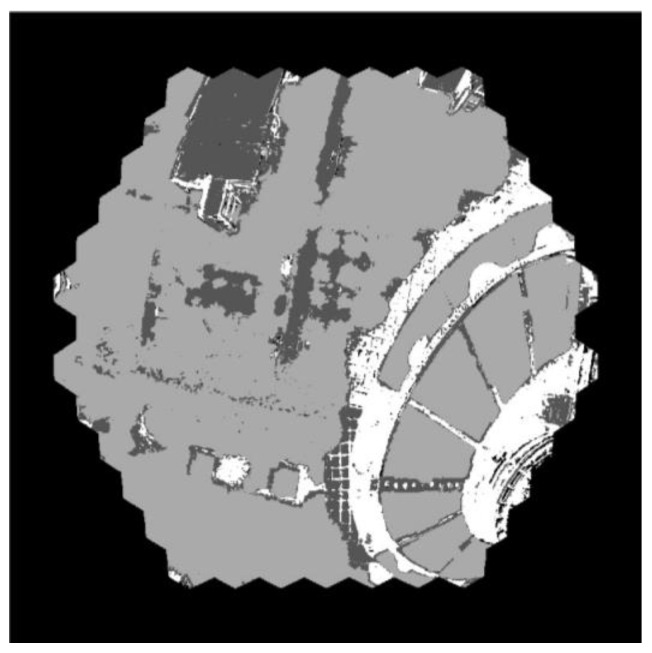
The object classification result with the k-means algorithm.

**Figure 18 biomimetics-08-00556-f018:**
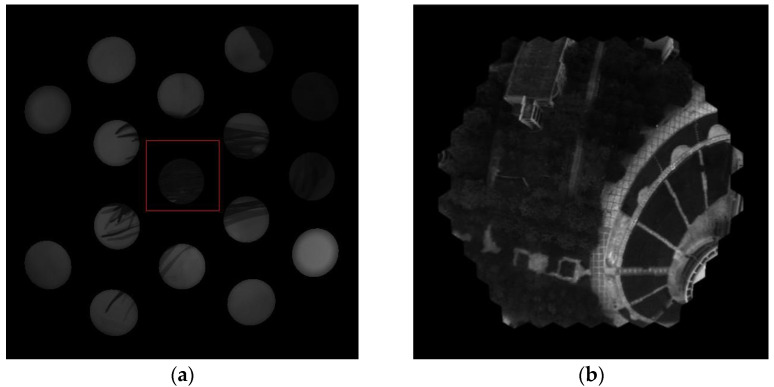
(**a**) The reconstruction image of BMCCEC, (**b**) the reconstruction image of BM3C.

**Table 1 biomimetics-08-00556-t001:** Key parameters of the BM3C prototype.

Parameter	Designed Value	Tested Value
Number of ommatidia	169	169
System focal length (mm)	2.7	2.76
Maximum FOV (degrees)	120	127.4
Central wavelengths (nm)	500, 560, 600, 650, 700, 750, 800	506, 560, 602, 653, 704, 751, 801
Spectral resolution (nm)	10	11.6
Maximum framerate (fps)	13	13
Distortion	<2%	<1.78%

**Table 2 biomimetics-08-00556-t002:** The central wavelength calibration result.

Ideal Wavelength	Ring 1	Ring 2	Ring 3	Ring 4	Ring 5
500	507.5	507.1	506.5	505.5	504.4
560	561.7	561.3	560.5	559.3	557.9
600	603.6	603.0	602.1	600.7	599.1
650	654.9	654.3	653.3	651.7	649.9
700	705.9	705.4	704.3	702.9	701.1
750	752.9	752.3	751.3	749.7	747.9
800	803.5	802.9	801.7	800.1	798.1

**Table 3 biomimetics-08-00556-t003:** The parameter comparison of aerial multispectral cameras.

Product Model	Size/mm	Weight/g	Bands	Wavelengths/ nm	Resolution/ Pixels	FOV/Degrees	Power/W
Micasense RedEdge	120 × 70 × 50	180	5	475, 560, 668, 717, 840	1280 × 960	47.2 × 35.4	4
Parrot Sequoia	59 × 41 × 28	135	4	550, 660, 735, 790	1280 × 960	61.9 × 48.5	8
Tetracam MCA6	116 × 80 × 68	580	6	490, 550, 680, 720, 800, 900	1280 × 1024	38.3 × 31.0	9.8
ADC lite	114 × 77 × 61	200	3	560, 660, 840	2048 × 1536	44.5 × 34.8	2
BM3C	194 × 194 × 232	2492	7	506, 560, 602, 653, 704, 751, 801	5120 × 5120	Max. 127.4	6

**Table 4 biomimetics-08-00556-t004:** The comparison of the optical parameters of multispectral compound eye systems.

System	Focal Length (mm)	FOV (Degree)	Wavebands	Number of Units
TOMBO (2004) [29]	1.3	/	7	16
TOMBO (2010) [19]	2.35	16 × 16	2	25
Planar multispectral ACE [30]	/	/	4	36
Multi-layer compound eye [31]	/	/	4	12
TOMBO (2020) [23]	1.5	50 × 50	8	9
BM3C	2.76	Max.127.4	7	169

## Data Availability

The data presented in this study are available on request from the corresponding author. The data are not publicly available due to funders’ policy.

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
