# Peer review of "Advanced Biomimetic Multispectral Curved Compound Eye Camera for Aerial Multispectral Imaging in a Large Field of View"

_biomimetics, 2023, doi:10.3390/biomimetics8070556_

Round 1
Reviewer 1 Report
Comments and Suggestions for Authors
The authors present a very interesting solution based on the idea of complex eye, which was proposed in research, running for about twenty years. The solution described in the manuscript is original and fits into the recent development of this subject, especially usefull for surface monitoring using drones. The result of the work shown in Fig. 4 is excellent.
I found some ambiguities in the text. In lines 66-70, the authors declare that they have received a system in which Blind Areas problems were overcomed, while several lines below (76-77) write that the image reconstruction leads to Blind Areas - so something is not agreed here; Perhaps this is just a language error - it needs to be explained and improved.
Further, I am not satisfied with the wording in line 99; The interefence occurs usually between signals/waves and not between the mounting elements of the filters - so is the problem of mechanical assembly or the phenomenon of interference? Please, write it more clearly.
And know, that figs. 3b-3c were generated by commercial software, however there is no possibility to precisely comment the results, since the resolution is not acceptable. Maybe it could be improved somehow using the appropriate graphic tool?
In what units are values on the axes Fig. 5b?
Descriptions in Fig. 6a and 6b are invisible.
In many places, the article requires a typical editorial correction; checking commas, spaces, font style, etc.
In conclusion, I find that the authors presented an innovative device by making an appropriate design analysis, which will definitely be commercially used. Due to the shortcomings I found, I suggest publication after making appropriate corrections, understood as Minor Changes.
Reviewer 2 Report
Comments and Suggestions for Authors
1. Please clarify the originality and novelty of your study compared to that of previous researches on the camera for aerial multispectral imaging.
2. Please compare the performance of the BM3C with that of the existing compound eye camera.
3. Please show the measured optical properties such as optical loss, transmittance, numerical aperture, and f number for each wavelength of BM3C and compare them with theoretical values.
4. Please explain why you chose the compound eye for the design of BM3C, because a single eye seems to be more suitable for application to drones.
5. Please use pictures of text or people in your experiment to make it easier to evaluate the results (e.g. Fig. 14).
6. Please modify the graph to make it distinguishable even when printed in black and white (e.g. Fig. 15).
Comments on the Quality of English LanguageMinor editing of English language required
Reviewer 3 Report
Comments and Suggestions for Authors
This manuscript discusses the development of compound eye cameras possessing multispectral imaging features. Drawing inspiration from the natural compound eye’s geometry, the creators have engineered imaging systems that boast a wide field of view (~120 degrees), incorporating 169 ommatidia. These cameras, equipped with multispectral filters across each ommatidium, have the capacity to capture images across seven distinct spectral bands. The acquired measurements fall within acceptable ranges, and the manuscript is articulately composed. Here are a few suggestions to enhance the manuscript’s quality:
-
The choice of wavelengths ranging from 500 nm to 800 nm raises curiosity. It would be beneficial if the authors could clarify the reason behind this selection, emphasizing the inclusion of the blue color spectrum within the visible range.
-
With regards to the organization of the multispectral filter, how would the arrangement be modified if the number of wavelengths was increased from seven to eight? It would be insightful to understand the positioning adjustments for each ommatidium in this scenario.
-
Consider elaborating on the optimization process between the quantity of wavelengths and spatial resolution. An exploration of the trade-offs involved would be quite instructive.
-
The introduction could be enriched by delving deeper into the concept of bio-inspired compound eye imaging systems. Providing a more comprehensive explanation, supplemented by relevant references (e.g., https://onlinelibrary.wiley.com/doi/10.1002/adfm.201705202), would facilitate a better grasp of this subject.
-
Compared to other systems, the present version appears somewhat cumbersome and voluminous. Could you outline any strategies in place to streamline and reduce the size of this model?
Minor editing is required.
Round 2
Reviewer 2 Report
Comments and Suggestions for Authors
1. The previous review comment #1 is asking about the originality and novelty of the ideas not asking about the performance comparison. It is required to clarify the main idea of this study compared to existing studies.
2. The performance of the compound eyes in Table 1 such as FOV depends on the number of units. Please explain the key ideas and scientific reasons for improving the performance of BM3C listed in Table 4 compared to other systems.
